# Microplastics in the Food Chain

**DOI:** 10.3390/life11121349

**Published:** 2021-12-06

**Authors:** Klára Cverenkárová, Martina Valachovičová, Tomáš Mackuľak, Lukáš Žemlička, Lucia Bírošová

**Affiliations:** 1Institute of Food Science and Nutrition, Faculty of Chemical and Food Technology, Slovak University of Technology, 81237 Bratislava, Slovakia; klara.cverenkarova@stuba.sk (K.C.); lukas.zemlicka@stuba.sk (L.Ž.); lucia.birosova@stuba.sk (L.B.); 2Faculty of Nursing and Professional Health Studies, Slovak Medical University, 83303 Bratislava, Slovakia; 3Institute of Chemical and Environmental Engineering, Faculty of Chemical and Food Technology, Slovak University of Technology, 81237 Bratislava, Slovakia; tomas.mackulak@stuba.sk

**Keywords:** microplastics, food chain, organic pollutants, microorganisms, health

## Abstract

Currently, microplastics represent a widespread contamination found in almost every part of the environment. The plastic industry has generated waste since the 1950s, which unfortunately now counts in the millions. The largest share of plastic consumption is used to produce packaging materials, including those applied in the food industry. The versatility of plastic materials is mainly due to their lightness, flexibility, strength, and persistence. Although plastic materials are widely used due to their beneficial properties, contamination of the environment with microplastics and nanoplastics is an emerging problem worldwide. This type of contamination is endangering animal life and thus also the food chain and public health. This review summarizes the knowledge about microplastics in the food chain. The effect of microplastics on the food chain has been particularly studied in marine organisms, and research deals less with other food commodities. Therefore, based on the studied literature, we can conclude that the issue is still not sufficiently examined, and should be paid more attention to maintain the health of the population.

## 1. Introduction

The industrial production of plastics dates back to 1950, when the annual production of plastics was at the level of 2 million tonnes. By 2015, global plastics production had grown to 380 million tonnes per year. Between 1950 and 2015, more than 7800 million tonnes of plastic were produced, generating 6300 million tonnes of waste, of which approximately 9% was recycled. Twelve percent of this waste was incinerated and the remaining 79% of the waste ended up in landfills or in the environment [1]. Much of the plastic waste ends up in the aquatic environment. It enters there not only directly, but also from landfills, from which plastics are spread by wind, into rivers, and then into the seas [2]. It is estimated that 8 million tonnes of plastics enter the seas and oceans each year [3].

Recently, microscopic plastic particles have come to the forefront of scientific interest. Plastic waste in the form of small pellets was first described in 1972. This type of waste was present in the Sargasso Sea [4]. In 2009, the U.S. National Oceanic and Atmospheric Administration (NOAA) defined particles as small as 5 mm as microplastics [5]. However, the lower limit of microplastic size is still under discussion and there is no general agreement on it. For example, according to Crawford and Quinn, microplastics are particles with a size of 5 mm to 1 μm, with particles in the range of 1 mm–1 μm being referred to as mini-microplastics [6]. Microplastics can be divided into two main groups in terms of origin, namely primary and secondary microplastics. A significant difference between the two types of microplastics is in the way that they enter the environment. Primary microplastics are released into the environment in their final form, while secondary microplastics are formed by weathering and wear of larger plastics into smaller particles directly in the environment [7].

A large increase in the use of microplastics has been observed mainly in cosmetics, as a patent has allowed the use of microbeads in exfoliating products since 1980. Until then, ground almonds, oatmeal, or pumice were used in these types of products. The presence of these particles in the formulation usually indicates the use of the words “microbeads” or “microexfoliants” on the package [8].

One of the largest sources of primary microplastics is the raw material used for the production of plastic products. In 2013, the total EU consumption of these particles was 53 million tonnes. The largest share, 40%, was used to produce packaging materials, while 20% of the plastics produced were used in construction. Environmental pollution occurs with the loss of pellets, which far exceeds the importance of microplastic pollution from cosmetics [9,10].

Secondary microplastics are irregular pieces of plastic that are created by unintentional degradation of large plastic objects, such as plastic bags, boxes, ropes, and nets. Over time, these large pieces of plastic break down into smaller and smaller particles by ultraviolet radiation from the sun and by mechanical influences, such as waves [6]. The release of secondary microplastics into the environment happens by three mechanisms:Natural disintegration of microplastics by weathering and microbial activity;Decomposition of macroplastics into microplastics by direct activity of organisms;Resuspension of past microplastic contamination in soil or sediment [11].

The microscopic dimensions and physical properties of these particles predetermine their easy spread into environments affected by human activity, but also into remote areas. It was only a matter of time before the presence of microplastics was detected in the human and animal food chain.

The main goal of this review is to summarize the occurrence of microplastics in the human food chain, with emphasis on the role of microplastics as vectors of various organic micropollutants and microorganisms. We will also mention the health consequences associated with the consumption of microplastics.

## 2. Synthetic Polymers as a Source of Microplastics

Over the decades, synthetic polymers have become a common part of our lives, and many natural materials have been replaced by these polymers. Their outstanding properties, such as variability, lightness, flexibility, strength, and persistence, are responsible for the versatile use of plastic materials. As all these properties are suitable for packaging food and other goods, more than a third of the plastics produced are intended for the packaging industry. Several types of plastics are used for packaging, most often polyethylene (PE), polypropylene (PP), polystyrene (PS), polyvinyl chloride (PVC), and polyethylene terephthalate (PET) [12]. Therefore, it is no surprise that with the growing volume of packaging plastics, a large amount of waste is generated which then ends up in the environment and in the oceans [13].

Plastic debris in the ocean comes largely from land waste (80%). The remaining 20% of the waste comes from sources in the ocean, mostly from fishing and trawls [14]. Fishing waste is estimated to account for 18% of all marine waste. It arises mainly from crumbling nets, ropes, and abandoned vessels [15]. The most commonly plastics used are polyolefins (PE and PP) and nylon [6,13]. The particular types of polymers that form microplastics in the oceans mimic the volume of plastic polymers produced. Polyethylene, polypropylene, polystyrene, polyester (PES), polyvinyl chloride (PVC), polyamide (PA), and polyvinyl acetate (PVA) microplastics are the most common in marine waters and sediments. However, the most dominant are PE, PP, and PS microplastics [16].

## 3. Characterization of Microplastics

The size, shape, and color of microplastics in an aqueous environment can vary. The basic difference is between primary and secondary microplastics. Primary microplastics, as industrially produced particles, have a regular, usually spherical, or fibrous shape and their surface is uniform; secondary microplastics, as weathering and erosion products, more often have an indeterminate shape. However, it is true that both types of microplastics can be significantly altered by weathering [6].

The size of the microplastics obtained from water or sediment depends on the sampling strategies and processing of the sample. According to a review by Hidalgo-Ruz et al. comparing the methodology for identification and quantification of microplastics from the marine environment in 68 studies, particles >500 µm are retained in standard sieves and can be sorted using a dissecting (stereo) microscope, while particles <500 µm are usually only obtained by studies using density separation and filtration. The authors also recommended dividing the obtained microplastics into size categories of 5 mm–500 µm and <500 µm [17]. The capture of the smallest fractions of microplastics is most problematic, as extraction losses and errors in distinguishing microplastics from particles of natural origin can easily occur [16]. The size of the microplastics is also crucial in their progress in the food chain of animals, as smaller microplastics are more accessible, e.g., for plankton [18].

The most commonly reported types of microplastics are pellets, fragments, and fibers, with films, ropes, filaments, sponges, foams, rubber, and microbeads (in decreasing order) also being important contributors to microplastic pollution [19]. The shape of the microplastics is an important indication of their origin, as fragments are usually formed by the breakdown of other plastics or fibers. According to the literature, fibers and fragments are most common in seawater [16]. Pellets can have tablet-like, oblong, cylindrical shapes, mostly spherical to ovoid with rounded ends, while fragments with sharp edges indicate a recent introduction into the sea or the recent break-up of larger pieces. Smooth edges are often associated with older fragments that have been continuously polished by other particles or sediment [17].

Just as microplastics can vary in size and shape, they can also be found in a wide variety of colors. The most common colors are black, blue, white, transparent, red, and green. A specific case is the “multicolor” hue, as some microplastics may have different colors on different parts. In addition to these, colors such as purple, yellow, brown, pink, and others occur to a lesser extent [20]. Color is considered important for studies concerning aquatic organisms, as some species are thought to potentially ingest microplastics based on a color preference behavior. In addition, color can also indicate the extent to which microplastics are contaminated with pollutants. Yellow and black microplastics are the most contaminated by persistent organic pollutants [21,22,23,24], while transparent and white microplastics are most often swallowed by marine animals [25]. According to Hidalgo-Ruz et al., white, pale yellow, and cream microplastics are most often recorded in the literature [17]. The color of the microplastics can also be used to estimate the type of polymer, as clean and transparent microplastics are often made of PP, while white and matte microplastics are often made of of PE and low-density polyethylene (LDPE). The degree of weathering of the microplastics can also be estimated according to the wear and fading of the surface [26].

The type of polymer, size, shape, and color provide comprehensive data on the identified microplastics. Although there are no official standards for the classification of microplastics at this moment, several manuals have been developed for this purpose which seek to facilitate their characterization [6,20,26].

## 4. Potentially Toxic Additives Released from Microplastics

In the first studies dealing with the issue of microplastics, the authors already assumed that various pollutants can be trapped on their surface. According to Carpenter and Smith, plastic particles in the Sargasso Sea could have been a source of polychlorinated biphenyls (PCBs), while in another study from the same year they confirmed that bacteria and PCBs at a concentration of 5 ppm were present on the surface of PS microbeads from seawater [4,27]. In the 21st century, research in this area has advanced, and the first conclusions have been drawn that state that microplastics can serve as a carrier for the spread of toxic chemicals in the marine environment.

Besides PCBs, organochlorine compounds, polyaromatic hydrocarbons, DDT and HCH insecticides, heavy metals such as copper, arsenic, cadmium, lead and chromium, and antibiotics can contaminate microplastics [28]. Microplastics with adsorbed pollutants can pose a potential risk to marine organisms, especially when entering the food chain by ingestion [29]. The concentration of chemical pollutants on microplastics can be one hundred to a million times higher than in the surrounding water [30]. The ingestion of microplastics with adsorbed micropollutants by aquatic animals is a way in which these toxic pollutants enter organisms [25]. Some studies consider organic pollutants that have accumulated on microplastics to be minor compared to environmental sources; however, the concentration of microplastics in the environment is constantly increasing and changing, which may result in temporary hot spots of microplastics as vectors of organic pollutants [31].

The incorrect indication and excessive use of antibiotics in human and veterinary medicine has led to problems beginning shortly after their massive spread. One of the common issues has been the insufficient removal and degradation of active compounds or their residues, which have subsequently been released into the environment. As a result, they are currently classified as pollutants. Trace amounts of antibiotics have been recorded in surface water, groundwater, and marine waters. In addition, there is no complete elimination of these substances in conventional wastewater treatment plants (WWTPs) [32]. This unfavorable situation was also documented in Slovakia. After the complete process of water treatment in the Bratislava WWTP, 12 types of antibiotics were removed, but the concentrations of azithromycin and clarithromycin in the effluent were still high (919 and 684 ng.L^−1^, respectively) [33]. Antibiotic residues in the environment create a selective pressure on microorganisms, which can lead to the advantage of resistant strains [34]. Antibiotics enter the environment from pharmaceutical production, wastewater and sludge, and often from improper disposal of these drugs by landfilling [35].

In a laboratory experiment, Li et al. incubated commercial microplastics in a concentration gradient of five antibiotics in seawater or plain water. They found that polyamide microplastics adsorbed four antibiotics to a greater extent (ciprofloxacin, amoxicillin, trimethoprim, and tetracycline), and that other microplastics also showed better adsorption in this water [36]. In another similar experiment, it was found that the antibiotics azithromycin and clarithromycin were able to bind to microplastics to a large extent, with the best binding capacity to PS. During the study, it was also observed that antibiotic-loaded microplastics significantly inhibited the growth and chlorophyll content of the cyanobacterium *Anabaena* sp., while virgin microplastics were not toxic to cyanobacterium. Most of the sorbed antibiotics were released upon contact with the cyanobacterial cultures, which was the cause for the observed toxicity [37].

The attachment of organic micropollutants to the surface of microplastics takes place by sorption processes, which include absorption and adsorption. Adsorption involves the entrapment of a substance on the surface of a particle, and upon absorption, the substance is assimilated into the mass of the particle [38,39]. Sorption processes are mediated by various interactions, most commonly through hydrophobic (non-polar matter vs. non-polar particle surface) and electrostatic bonds (attracting oppositely charged/repelling equally charged molecules) [38].

Ionic, van der Waals, steric, and covalent bonds are applied in the adsorption process, while only the weaker van der Waals forces are applied in absorption [40]. The binding of organic micropollutants to the surface of microplastics depends on several factors, such as the type of polymer, color, size, and degree of weathering of the microplastics, as well as on pH, salinity, and seawater temperature [41].

Microplastics can not only sorb and bind toxic pollutants from the environment, but they can often release these compounds back. In the production of plastics, various additives (stabilizers, plasticizers, and flame retardants) that are used as processing compounds in relatively high concentrations can later be released during their disintegration [13,29]. The most commonly used additives are brominated flame retardants (BFRs), phthalates, nonylphenols, bisphenol A, and various antioxidants. All of these substances have either a potential or proven effect as endocrine disruptors. Leaching of hazardous substances from plastic waste is important especially for flame retardants and phthalates, as they are not part of the polymer matrix. Bisphenol A and antioxidants are especially dangerous in the food industry, as they are used in packaging materials in contact with food [42]. Leaching of additives from plastic particles is mainly determined by the partition coefficient (K_PW_) between water and plastic, or another type of partition coefficient [43]. Desorption by the digestive system is particularly important when microplastics are swallowed by animals [44]. Koelmans et al. assessed the potential of leaching of bisphenol A and nonylphenol from swallowed microplastics in the intestinal tracts of lugworm and cod. They found that plastic ingestion by the lugworm is a significant route of exposure, but the risk is reduced due to the low environmental concentrations of nonylphenol and bisphenol A. For cod, this route of exposure was marginal [45]. However, these results do not indicate that the contribution of additives released from microplastics is negligible. In the open sea, the amount of additives released from microplastics may be low, but their accumulation in enclosed and semi-enclosed bays and sediments needs to be investigated [44].

## 5. Interaction of Microorganisms and Microplastics

Microorganisms are found in large numbers in the marine and aquatic ecosystem. Marine sediment is a natural habitat for many microorganisms. These microorganisms are involved in the nutrient cycle in water [46]. Therefore, it is not surprising that with the degree of plastic pollution of marine and freshwater resources, the first information about microbial association with microplastics soon appeared. The groundbreaking article by Carpenter and Smith stated that hydroids and diatoms were present on the surface of plastics in the Sargasso Sea [4]. Filamentous fungi, algae, and, above all, bacteria have been found on the surface of microplastics [28]. In the first study aimed at the identification and characteristics of microbial populations on sea microplastics, Zettler et al. found that species from the genus *Vibrio* colonized the surface of the PP microplastic particle (the species was not specified). They hypothesized that plastics could then act as vectors of pathogenic microorganisms that can enter the digestive tract of fish and birds after swallowing. The term “plastisphere”, referring to microbial communities on microplastics, also appeared in this publication [47]. McCormick et al. identified bacteria of the *Pseudomonadaceae*, *Proteobacteria*, and *Campylobacteraceae* families on microplastics from the Chicago River, with pseudomonads accounting for up to 19% of all 16S rRNA sequences on the microplastics [48].

The formation of a microbial biofilm is a very common method of colonization on the surface of microplastics. In a biofilm, microorganisms produce an extracellular polymer matrix, which protects them from external influences [28]. Microbes also bind to plastic particles due to the good availability of nutrients that are attached to the surface. Biofilms are hot spots for microbial competition and horizontal gene transfer. Adsorption of pollutants, such as antibiotics, on the surface of microplastics can create conditions that support this process [49,50]. Biofilms usually consist of bacteria (mainly *α*-*Proteobacteria*) and diatoms and, in the case of marine biofilms, may also contain pathogenic microorganisms [49,51]. Biofilm formation on marine microplastics usually begins with *γ-Proteobacteria* (*Pseudomonas*, *Alteromonas*) in the first 24 h and continues with *α-Proteobacteria* after 24 h. Over time, bacteria from the *Bacteroidetes* phylum are attached [49]. It is not uncommon for the composition of microbial biofilms on microplastics to differ significantly from microorganisms present in the surrounding water or biological material [47,52]. Oberbeckmann et al. found that the composition of biofilms on microplastics may vary depending on the geographical location, season, and type of polymer. Bacterial phyla *Bacteroidetes*, *Cyanobacteria*, and *Proteobacteria* and eukaryotes *Stramenopiles* were the most identified microorganisms on microplastics [52].

The presence of a biofilm on the surface of the microplastic can lead to its alteration and thus facilitate microplastic consumption by aquatic organisms [53]. After consumption and passage of the microplastic through the intestinal tract of an aquatic organism, bacterial strains which form its microbiota can be attached on the surface before the microplastic is further excreted into the environment. There it can be swallowed by another organism in the next trophic level [49]. As the presence of pathogenic microorganisms, such as *Vibrio*, have been shown on microplastics, they can serve as vectors of these pathogens to organisms; however, the literature also suggests other routes of transmission, such as contact with skin or mucous membranes (e.g., when children play on the beach in the sand) [25]. In addition to pathogenic species of the genus *Vibrio*, the presence of other pathogens has been confirmed on microplastics. *Escherichia coli* strains, which according to PCR analysis belonged to virulent enteropathogenic serovars, were isolated on microplastics from Guanabaro Bay, Brazil. Strains of *Vibrio* spp. were isolated as well, specifically *Vibrio mimicus*, *Vibrio vulnificus*, and *Vibrio cholerae* [54]. The invasive fish pathogen *Aeromonas salmonicida* was identified on microplastic fragments from the Slovenian Adriatic coast, together with the pathogens *Acinetobacter* and *Haemophilus* [55]. Opportunistic human pathogens *Pseudomonas monteilii* and *Pseudomonas mendocina*, and the plant pathogen *Pseudomonas syringae* were identified in a biofilm formed on plastic particles in river water [56]. It is hypothesized that microplastics may be carriers not only of a wide variety of micropollutants but also of pathogenic microorganisms that are unable to spread independently [57,58]. Another problem is the fact that microplastics with a biofilm can be involved in horizontal gene transfer (HGT) between different bacteria, thus promoting the transfer of antibiotic resistance [59]. Therefore, microplastics are a hot spot for organic micropollutants, mobile genetic elements, and microorganisms (Figure 1).

## 6. Sampling and Analysis of Microplastics from Different Matrices

Research has currently confirmed the penetration of microplastics into global aquatic ecosystems as well as the food chain [60]. Given the determination of the real risk that microplastics and nanoplastics pose, it is necessary to develop and implement standard protocols for the collection, quantification, and characterization of microplastics. Analytical techniques include sampling, sample preparation, identification, and quantification of microplastics or nanoplastics. Despite the great interest in the topic of micro- and nanoplastics, there is still a lack of standardized procedures in the research area for their efficient extraction, especially from sediment, air, or biological tissues [61]. There is also a lack of procedures for analysis of nanoplastics [61].

Microplastics can be found in various types of samples. There are several ways to take samples depending on their location [61]. For environmental samples (especially sediment and water), collectors with a rectangular entrance and a collection bag made of mesh are used. Conical bongo nets are used for sampling from the middle part of the water surface. Samples of marine or river sediment are taken using metal spoons or nets. The mesh size of the nets may vary from 53 µm to 3 mm, which in turn affects the nature of the sample taken. Samples are taken from biological tissues primarily by dissection of animals, or after spontaneous leakage of the sample from the digestive system [62].

The collection of microplastics from biological samples depends mainly on the size of the organism. Chemical or enzymatic decomposition of organic material is used to separate microplastics. In particular, nitric acid, nitric acid:hydrochloric acid in a ratio of 3:1 by volume, hydrogen peroxide, and nitric acid:perchloric acid in a ratio of 1:1 by volume are used [61,63]. In addition, bases (sodium hydroxide, potassium hydroxide) and relevant enzymes (proteinase-K, lipase, cellulase, chitinase) are applied for decomposition of organic matter [63,64,65]. A problem with the application of chemicals may be their effect on smaller microplastics, not just in the form of fibers. Chemicals can degrade certain groups of plastics. Therefore, the use of enzymes, such as proteinase, lipase, cellulase, or chitinase, seems to be a more suitable option for sample preparation [66].

Another important step in the analysis of microplastics is the cleaning of their surface. This procedure often overlaps with the isolation of the sample when chemicals (HCl, NaOH, H_2_O_2_) or enzymes are used, or when ultrasound is applied [67]. The reason for surface cleaning is the removal of solid impurities or biological contamination clogging the sample surface. However, if surface cleaning is not required (i.e., the sample can be analyzed without this pre-treatment), this step can be skipped as it may affect the decomposition of some types of plastics [61].

Raman spectroscopy and Fourier-transform infrared spectroscopy (FTIR) are currently the most widely used analytical methods for studying microplastics. Samples larger than 1 μm can be analyzed by Raman spectroscopy. Compared to FTIR spectroscopy, Raman spectroscopy achieves a more sensitive response to nonpolar symmetric bonds, whereas FTIR is more sensitive to the identification of polar groups [68]. Pyrolysis combined with gas chromatography and mass spectroscopy (py-GC-MS) may be used for the characterization of certain polymers based on their degradation products. With this methodology, it is possible to simultaneously identify the type of polymer and the organic filler. However, this is a destructive method [61,69]. In addition, it is not possible to distinguish LDPE from high-density PE (HDPE) using py-GC-MS [61].

Using nuclear magnetic resonance (NMR) spectroscopy, it is possible to obtain information on the chemical structure of the polymer chain of microplastics, detailed information on monomers in the case of copolymer compounds, the degree of crystallinity in semi-crystalline polymers, and information on branching and tacticity [61]. For complex analysis of microplastics, new technologies are constantly being developed that differ primarily in the method of sample preparation. One of these new methods is extraction with compressed liquid [70]. Another developing methodology is the quantification of microplastics based on selective fluorescence labeling using the lipophilic dye Nile Red [71]. A major challenge is the analysis of nanoplastics below 100 nm. Scanning electron microscopy (SEM) techniques are mainly used here [72].

For the analysis of sorbed chemicals on the surface of microplastics, classical extraction procedures and LC-MS/MS or GC-MS techniques are used. To remove the metals adsorbed on microplastics, the sample is extracted with a 20% solution of HCl:HNO_3_ (3:1). The metals are then analyzed by inductively coupled plasma mass spectroscopy (ICP-MS) or atomic absorption spectroscopy (AAS) [61,73]. Other analytical procedures for the analysis of chemical compounds accumulated in microplastics are gas chromatography with electron capture detector (GC-ECD), gas chromatography-mass spectrometry (GC-MS), X-ray fluorescence (XRF), or SEM with energy-dispersive X-ray spectroscopy (SEM-EDS) [61]. There is no standard protocol for the identification and quantification of microplastics in different types of samples, which has a negative impact on the interlaboratory comparison of results [61].

## 7. Microplastic Contamination of the Food Chain

As microplastics contaminate the environment, their presence has been demonstrated in the food chain. At lower trophic levels in the marine environment, the presence of microplastics has been reported in zooplankton, chaetognatha, ichtyoplankton, copepods, and salps. Microplastic contamination also occurs at higher trophic levels, in invertebrates (polychaetes, crustaceans, echinoderms, bivalves) and vertebrates (fish, seabirds, and mammals). Plastic particles reach them either through direct consumption or through trophic transfer (Figure 2) [74].

The bioaccumulation of microplastics in the digestive tract of fish is not of great concern for humans, as this part is usually not consumed. More serious is the bioaccumulation in crustaceans, which are filter feeders and their digestive tract is consumed [75]. The level of pollution in seafood and organisms intended for human consumption can be detected by research into indicator species from the marine environment. Mussels and mollusks are a good indicator as they are consumed whole and can be a significant source of microplastics. Benthic fish may indicate sediment contamination. Sardines and anchovies are also consumed whole, and the number of microplastics in them speaks of contamination of the open sea and subsequent human exposure [76]. Van Cauwenberghe and Janssen determined the microplastic content in the soft tissue of two commercially grown mollusks (blue mussel *Mytilus edulis* and Japanese oyster *Crassostrea gigas*). The amount of microplastics was 0.36 ± 0.07 particles.g^−1^ ww (wet weight) in the mussels and 0.47 ± 0.16 particles.g^−1^ ww in the Japanese oysters, with the average European consuming up to 11,000 microplastics per year [77]. Microplastics can also be found in canned products, such as sardines and sprats. In a study from Karami et al., the presence of micro- and mesoplastics was confirmed in 20 brands of canned sardines and four brands of canned sprats [78].

Microplastic contamination occurs not only in seafood but also in other foods. Liebezeit and Liebezeit identified colored and transparent fibers and fragments in honey samples (mainly from Germany). The fiber numbers ranged from 40 to 660 particles.kg^−1^ of honey (mean 166 ± 147 particles.kg^−1^). Fragment numbers were lower (0–38 particles.kg^−1^ honey; mean 9 ± 9 particles.kg^−1^ honey). Next, the presence of microplastics in sugar samples was investigated, where fibers and fragments of 217 ± 123 particles.kg^−1^ of sugar and 32 ± 7 particles.kg^−1^ of sugar, respectively, were identified [79]. The same authors analyzed 24 samples of German beers in a later study. They most often identified the presence of fragments. Fibers and granules were less common. The numbers found ranged from 2–79 fibers.L^−1^, 12–109 fragments.L^−1^, and 2–66 granules.L^−1^ [80]. 

A Chinese study mapped the occurrence of microplastics in sea salt, as the authors assumed their presence was due to production from seawater. There were 550–681 particles.kg^−1^ in sea salt, 43–364 particles.kg^−1^ in lake salt, and 7–204 particles.kg^−1^ in rock salt. The particles consisted mainly of fibers and fragments, and more than half of the detected microplastics had a size below 200 µm [81].

The occurrence of microplastics was further recorded in bottled water. In Germany, bottled water in disposable and returnable plastic bottles and in beverage cartons was tested for the presence of microplastics. Disposable bottles contained 14 ± 14 particles.L^−1^ and returnables contained 118 ± 88 particles.L^−1^, which is up to 8 times more (compared to beverage cartons, it is up to 10 times more). Higher numbers of microplastics were also found in some glass bottles. The microplastic material was related to the bottle material [82]. Mason et al. found an average of 10.4 particles.L^−1^ in 259 bottled water samples from around the world. These were particles with a size > 100 µm. Together with the particles with a size of 6.5–100 µm, the bottles contained 325 particles.L^−1^ of bottled water. Fragments and fibers were most often identified [83].

The least information is available on contamination of non-marine or aquatic species. Huerta Lwanga et al. describe in their study the contamination of chicken gizzards intended for human consumption. The gizzards contained an average of 10.2 ± 13.8 microplastic particles [84]. The gardens where these chickens lived were heavily contaminated with plastic waste; therefore, this research cannot be considered representative of real meat contamination [85]. 

Since these groundbreaking findings on food chain contamination, new research has emerged where authors assess the contamination of new commodities (Table 1). Research is still focused primarily on contamination of seafood and fish, but interest is also focused on foods of plant origin, such as seaweed and rice. At the same time, the contamination of vinegar, salt, and milk has been evaluated. From beverages, the area of interest has expanded to white wine, energy drinks, and soft drinks.

Besides the contamination of air and water, soil contamination is another possible source of microplastics in the food chain. Soil microplastic contamination occurs via several routes. These include landfills, soil treatment, use of sewage sludge for soil fertilization, irrigation with wastewater, use of compost and organic fertilizers, remnants of mulching foils, tire wear, and atmospheric gradient. The presence of microplastics in soil reduces its quality and indicates that further fragmentation will occur within it [99]. Precisely due to the application of mulching foils, sludge, and wastewater from WWTPs, microplastics accumulate in the surface layers of agriculturally cultivated soil [100].

It is estimated that the use of sludge in agriculture contributes to the addition of 125 to 850 tonnes of microplastics per million inhabitants into European soil each year [28]. Microplastics can affect soil density and porosity, which can affect water dynamics and soil aggregation. In addition, according to some research, microplastics in the soil affect the amount of carbon, nitrogen, and phosphorus, which can disrupt the nutrient cycle [101]. Plastic fragments can migrate to lower layers in the soil, and agricultural activities, such as plowing, support this process. Crop cultivation itself (the formation of the root system, the collection of crops such as potatoes and carrots), dry weather (the formation of cracks through which plastics move deeper), and the action of soil organisms also contribute to the spread of microplastics to the lower layers of the soil [102]. Accumulation and migration of micro- and nanoplastics was observed in a laboratory experiment with radish (*Raphanus sativus*) where acrylonitrile butadiene styrene powder was applied to the root system of the plant [103].

According to estimates based on 26 studies, the US population consumes 39,000 to 52,000 microplastic particles per year (depending on age and gender). This data was obtained by evaluating 15% of the caloric intake of Americans, which included the content of microplastics in seafood, honey, sugar, salt, alcohol, and bottled and tap water, as well as microplastic intake from the air [104].

## 8. Effect of Microplastics on Human Health

Microplastics enter the human food chain mainly from contaminated foods and can have a potential impact on human health. Inhalation of microplastics is another contamination route of the human body [105]. A minor source of microplastics in the human body is skin contact [106].

The effect of microplastics in the human gastrointestinal tract after consumption is relatively unexplained. It is assumed that after consumption, the largest fraction of the micro- and nanoplastics is excreted in the feces (>90%). Absorption of microplastics by the intestinal epithelium probably only occurs with microplastics up to 150 µm in size, as microplastics this size were present in lymph in mammalian studies. Exposure to these microplastics leads to systematic exposure, while larger microplastics can only produce local effects on the immune system (e.g., inflammation of the intestine). The smallest fraction of microplastics (<1.5 µm) can penetrate deep into the organs [107]. Nanoplastics pose a higher risk because their size allows them to cross the placenta and the blood-brain barrier, as well as transport across M-cells in Peyer’s patches in the small intestine to the blood and lymphatic system, from where they can contaminate the liver and gallbladder [108]. In a study from 2021, the presence of microplastics in human placenta was demonstrated for the first time. These were 12 fragments of 5 to 10 μm in size that were detected in four placentas. The method of their penetration into the placenta is still unknown, as well as possible effects on pregnancy and the fetus [109]. 

The transfer of microplastics into stool in humans has been reported by Schwabl et al. They identified 50 to 500 µm microplastics in stool samples from eight individuals. Fragments and fibers were the most common, and PP and PET predominated among the materials [110]. In another study, 23 of 24 stool samples from young Chinese men contained microplastics. Their size ranged from 20 to 800 µm, and the most common were PP, PET, and PS microplastics [111].

The long-term effects of microplastics on human health is relatively unknown [106,112,113]. Their adverse effects on the body may include the induction of oxidative stress by producing reactive oxygen species during the inflammatory reaction, which may lead to cytotoxic effects. Microplastic intake can upset energy balance, metabolism, and the immune system [113].

Another risk associated with the consumption of microplastics in food is microbial association with their surface. The presence of various pathogenic species has been confirmed on the surface of microplastics, and the consumption of seafood increases human exposure to these microorganisms. Harmful chemicals such as bisphenol A, PCBs, PAHs, chlorinated pesticides, BFRs, and antibiotics can be released from microplastics into food, which can subsequently have carcinogenic and mutagenic effects and act as endocrine disruptors [114]. However, according to some studies, persistent organic pollutants consumed with microplastics represent a negligible source of contamination for humans. In the case of bisphenol A, the estimated daily dose from a normal diet would be 40 million times higher than after eating contaminated seafood [115]. Further, the contribution to exposure to PCBs and PAHs from contaminated microplastics in mussels would be <0.006% and <0.004%, respectively [107].

During the global pandemic of COVID-19, wearing protective masks was one of the first measures to prevent the spread of the disease. These are often composed of synthetic polymers. However, disposable masks quickly became waste that polluted the environment and, in addition, could release fibrous microplastics [116]. In addition, the SARS-CoV-2 virus can survive on surfaces for up to 5 days, so scientists have begun to predict the spread of COVID-19 through microplastics released from used masks. At present, this possibility of transmitting the infection has not been confirmed or refuted [117,118].

## 9. Conclusions

Microplastics contaminate almost every part of the environment, including the food chain. They may adsorb different type of chemicals and microorganisms on their surface and thus increase contamination load. Since microplastics are relatively small, they are easily ingested and can adversely affect the consumers’ health. Although data about the occurrence of microplastics in the different stages of the food chain and in foodstuffs are growing, they are still insufficient. The biggest issue with determining microplastic contamination in food is the lack of a uniform methodology. The results of individual studies vary in the methodology used; therefore, the assessment of contamination is complex and difficult to interpret. A good step to reduce the microplastic load in the food chain and in the environment would be to introduce legislation regulating the use of primary microplastics and their release into the environment. The results presented here suggest that the effect of microplastics on the food chain, and especially the impact of microplastics on human health, need to be addressed much more intensively. A good tool to mitigate the potential negative impacts of microplastics in food would be risk analysis and the subsequent introduction of nutritional recommendations for high-risk foods with a higher microplastic content. Reducing the problem of microplastic pollution needs new technologies for their degradation in the environment. It is also very important to raise public awareness of microplastics and better waste management.

## Figures and Tables

**Figure 1 life-11-01349-f001:**
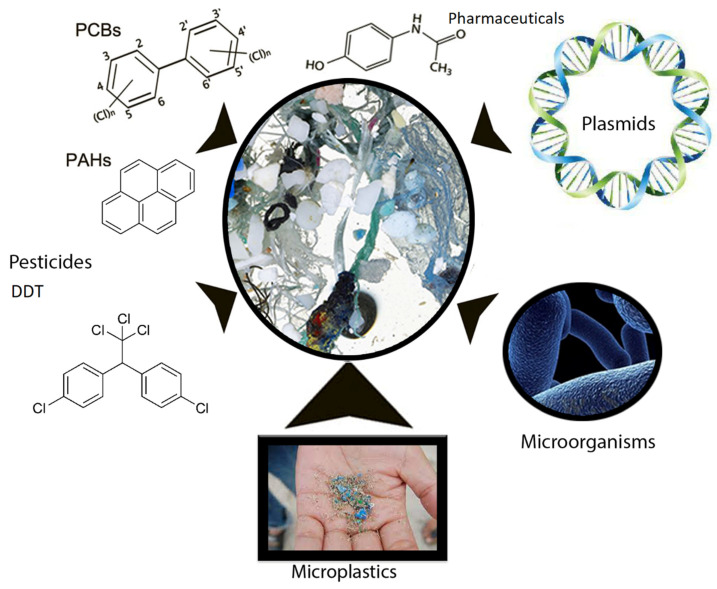
Microplastic particles can be carriers of various micropollutants and toxic metals as well as specific biological contaminants or resistance genes. (PCBs: polychlorinated biphenyls; PAHs: polycyclic aromatic hydrocarbons; DDT: dichlorodiphenyltrichloroethane).

**Figure 2 life-11-01349-f002:**
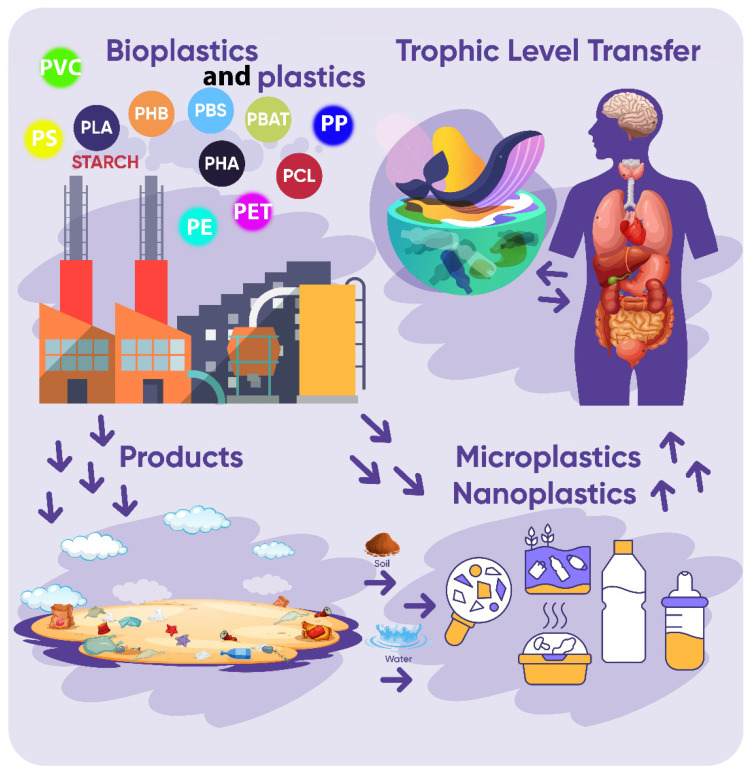
Lifecycle of microplastics in the environment and food chain. (PVC: polyvinyl chloride; PS: polystyrene; PLA: polylactic acid; PHB: polyhydroxybutyrate; PBS: polybutylene succinate; PBAT: polybutylene adipate terephthalate; PP: polypropylene; PHA: polyhydroxyalkanoates; PCL: polycaprolactone; PET: polyethylene terephthalate; PE: polyethylene).

**Table 1 life-11-01349-t001:** Occurrence and characteristics of microplastics in different food and drink commodities.

	Commodity	Location	Type of MP	Material of MP	Size Range	Level of MP	Reference
Seafood	Commercially important fish species (Australian herring, Australian salmon, Australian sardine, Australian snapper, dusky flathead, King George whiting, sea mullet, southern garfish, tiger flathead)	Australia	fibers, fragments, films	PE, PP, polyblends, acrylate, nylon, paint, PES, poly-vinyl	38 µm–>1 mm	0.96 ± 0.08 MP/fish	[86]
	Indian white shrimp (*Fenneropenaeus indicus*)	India	fibers, fragments, sheets	PA, PES, PE, PP	157–2785 µm	0.04 ± 0.07 MP.g^−1^ ww	[87]
	Golden anchovy (*Coilia dussumieri*)	India	fibers, films, fragments, pellets, beads	PE, PP, PA, PES, PS	<100–>1000 mm	6.78 ± 2.73 MP/fish	[88]
	Commercial seaweed nori	China	fibers, fragments, films, pellet	PES, rayon, PP, PA, cellophane	0.11–4.97 mm	1.8 ± 0.7 MP.g^−1^	[89]
Food	Chicken and turkey (packed in PS trays)	France	particles, fibers	extruded PS	300–450 µm	4.0–18.7 MP.kg^−1^	[90]
	Canned fish (mackerel and tuna)	Iran	fibers, fragments, films	PET, PS, PP, PS-PP, PS-PET, PVC, LDPE	fibers 100–8000 µm, fragments 10–1100 µm, films 70–1000 µm	1.28 ± 0.04 MP.g^−1^	[91]
	Uncooked rice	Australia	NR	PE, PP, PET	NR	67 ± 26 µg.g^−1^ dw	[92]
	Instant rice					283 ± 50 µg.g^−1^ dw	
	Table salts	Africa	microfibers, particles	polyvinyl acetate, PP, PE	3.3–4460 µm	38.42 ± 24.62 MP.kg^−1^	[93]
	Vinegar	Iran	fragments, fibers	PE, HDPE	1–500 µm (mainly)	51.35 ± 20.73 MP.L^−1^	[94]
	Milk	Mexico	fibers, fragments	Polyethersulfone, polysulfone	0.1–5 mm	6.5 ± 2.3 MP.L^−1^	[95]
Drinks	White wine	Italy	NR	PE	7–475 µm	2563–5857 suspected MP.L^−1^	[96]
	Tap water	Hong Kong	Fibers, films	NR	50–4830 µm	2.181 ± 0.165 MP.L^−1^	[97]
	Cold tea	Mexico	Fibers	PA, PEA	<1 mm	11 ± 5.26 MP/drink	[98]
	Soft drinks		Fibers	PA, PEA, acrylonitrile-butadiene-styrene	0.1–3 mm	40 ± 24.53 MP/drink	
	Energy drinks		Fibers	PA, PEA	<1 mm	14 ± 5.79 MP/drink	
	Beer		Fibers, fragments	PA, PEA, PET	<1 mm–2 mm	152 ± 50.97 MP/drink	

NR: not reported; PE: polyethylene; HDPE: high-density PE; LDPE: low-density PE; PP: polypropylene; PES: polyester; PA: polyamide; PS: polystyrene; PET: polyethylene terephthalate; PVC: polyvinylchloride; PEA: polyesteramide; MP: microplastics; ww: wet weight; dw: dry weight.

## Data Availability

Not applicable.

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
