# Peer review of "Microplastics in the Food Chain"

_life, 2021, doi:10.3390/life11121349_

Round 1
Reviewer 1 Report
Dear Authors,
The manuscript improved quite a bit from the previous version, and I can see most of the comments are integrated into this revised version.
There are only a few errors as follows:
L138 and L423 – Please, correct light-density polyethylene with low density polyethylene for LDPE
L284 – Please, add unsaturation to specific aromatic rings of PAHs (probably the pyrene structure is shown in Figure 1!)
L422 – Please, delete the comments/annotations from Table 1 (the column with references).
Author Response
We have incorporated all the changes requested by the opponent. We believe it will be fine.
Yours sincerely
Valachovičová

Reviewer 2 Report
The manuscript is revised according to the comments and it is now recommended for the publication.
Author Response
Thanks for the review.
Yours sincerely
Team of authors
This manuscript is a resubmission of an earlier submission. The following is a list of the peer review reports and author responses from that submission.
Round 1
Reviewer 1 Report
The title of this review "Microplastics in the food chain" is misleading. There is just a small paragraph on food. There are no tables and no figures that summarize the results and capture the reader's attention.
There are other articles and reviews about microplastics in the food chain that are more complete, for example: https://doi.org/10.1016/B978-0-12-813747-5.00011-4; https://doi.org/10.3390/ijerph17186710; https://doi.org/10.1016/j.chemosphere.2020.126787; https://doi.org/10.1007/s13197-019-04138-1; https://doi.org/10.3390/su12145514.
Recently some special issues like: "Microplastics in Aquatic Environments: occurrence, distribution and effects" and "Fate and effects of micro and nanoplastics in soil and aquatic ecosystems" have been published on the journal Toxics (MDPI). The presence of microplastics in seafood, honey, sugar, salt, meat, beer has been widely dicussed in these papers and elsewhere, so I don't think this review should be published.
Moreover, I don't think this topic is suitable for a journal like Life.
Reviewer 2 Report
The manuscript submitted to Life focuses on identifying microplastics (MPs) in the food chain, cosmetic products, or natural/wastewaters. Perhaps the authors were trying to describe, also, the primary sources of organic micropollutants through the leaching of plastic additives and the associated ecological/human risks. The article does deal with a new topic framed as a priority field of analysis. The work is generally well written but requires some corrections and changes in the structure of the paper:
- The authors should find more keywords that are not in the title of the manuscript
- Please, add the tables or figures to show the inventory of MPs by their types and functions and different plastics additives, respectively. All of them should be in correlation by the various food/cosmetics category. Also, add their potential health implication and references in the same tables.
- Please, discuss more adsorption mechanisms of organic micropollutants (OMs) on MPs. Summarize the toxicological interaction between MPs and OMs and human health, respectively.
- There is missing data about detection and identification qualitative/quantitative technical analysis of MPs in the different food products. There are no assessments about relations of MPs identification in food products by soil, for example.
And more specific remarks:
L142 – it is not written the chemical product corresponding to the LDPE acronym
L167,300, 301, 304 – it uses the symbol „L” for litter (U.M.)
L188 – for brominated flame retardants acronym is BFRs.
BFRs include PBDEs (polybrominated diphenyl ethers)
L357 – respectively at the final of the sentence.
I recommend preparing the references in according with the journal requirements. See, L392, 397-399, 403 etc.
The Conclusion part is straightforward and too general - more details are needed.
Reviewer 3 Report
Overall this article is interesting as it provides an overview of microplastics, their presence in the environment and food and the potential risks to human health
No information is given on how the literature search was conducted: over what period? How many articles were retained or excluded from the primary search? What key words were used to conduct the search? I feel that there is a lot missing
I am surprised that the authors do not mention the contamination of soils by plastics, especially through the use of plastic sails and materials in agriculture to protect crops. This pathway can also eventually contaminate food.
Line 54 the volume is expressed in tonnes.
Line 85 I don't understand why the word durability is used in this sentence another term if you want to make this material last over time
Line 136 give a reference to the fact that yellow or black plastics are the most contaminated by pollutants
What type of pollutants are we talking about?
Line 279 what is the unit for 11,000?
Reviewer 4 Report
The paper presents a review of the literature related to microplastics in the food chain. A significant number of data on different types of microplastics and their presence in the environment are included. However, I think it is necessary for the authors to make a few changes to make the text better in line with the title. The text dedicated to the description of different types of microplastics can be shortened. Interactions with antibiotics and microorganisms need to be placed in the context of the food chain. The Introduction does not define the goal of the paper. In conclusion, it is necessary to define knowledge gaps and future directions of research. Therefor I recommend minor revision.